*Brief communication:*

# Hurricane Dorian: automated near-real-time mapping of the "unprecedented" flooding on the Bahamas using SAR

Diego Cerrai[1], Qing Yang[2], Xinyi Shen[1], Marika Koukoula[1], Emmanouil N. Anagnostou[1]

[1]Department of Civil and Environmental Engineering, University of Connecticut, Storrs (CT), 06279, USA
[2]College of Civil Engineering and Architecture, Guangxi University, Nanning, Guangxi, 530004, China

*Correspondence to*: Diego Cerrai (diego.cerrai@uconn.edu)

**Abstract.** In this communication, we present application of the automated near-real-time (NRT) system called RAdar-Produced Inundation Diary (RAPID) to European Space Agency Sentinel-1 SAR images to produce flooding maps for Hurricane Dorian in the northern Bahamas. RAPID maps, released two days after the event, show that coastal flooding in the Bahamas reached areas located more than 10 km inland, covering more than 3,000 km$^2$ of continental area. RAPID flood estimates from subsequent SAR images show the recession of the flood across the islands, and present high agreement scores when compared to Copernicus Emergency Management Service (Copernicus EMS) estimates.

## 1 Introduction

Hurricane Dorian was the strongest Atlantic hurricane at landfall in terms of maximum sustained winds (185 mph, 83 ms$^{-1}$), tied with the 1935 Labor Day Hurricane (Landsea et al., 2014). Dorian's first record-breaking landfall occurred at 16:40 UTC and its second at 18:00 UTC on September 1, 2019, in the Abaco Islands in the northern Bahamas (NHC 2019). A third landfall occurred at 03:00 UTC on September 2 at the eastern end of Grand Bahama and was characterized by maximum sustained winds of 180 mph. Tropical storm conditions battered the northern Bahamas for 72 hours, and locations in northeastern Grand Bahama suffered hurricane conditions for more than 40 hours. Between 08:00 UTC on September 2 and 14:00 UTC on September 3, the National Hurricane Center issued 30 consecutive hourly public advisories (NHC 2019) indicating Hurricane Dorian was either moving at 1 mph or was stationary, resulting in prolonged extreme conditions over the same areas. The prolonged damaging and record-breaking winds were just one aspect of this storm - one that was measured directly and in near real time (NRT) by aircraft missions (HRD 2019). In addition, the combined effect of storm surge and heavy precipitation brought about the extensive flooding that was the major cause of "unprecedented and extensive devastation," as described by Bahamian Prime Minister Hubert Minnis. Neither precipitation nor coastal surge could be directly measured because of the lack of a ground-based observational network.

In this and other remote areas around the world where ground-based measurements are not available, precipitation can be assessed using near-real-time (NRT) satellite estimates available through the National Aeronautics and Space Administration (NASA) Global Precipitation Measurement (GPM) Integrated Multi-satellitE Retrievals for GPM (IMERG), version 06 (Huffman et al. 2019). Without these automated estimates, gaining prompt situational awareness becomes difficult, which can cause delays in rescue operations.

Several space missions acquire synthetic aperture radar (SAR) data derived from low Earth observation (LEO) satellites that can be used for NRT systems: Sentinel-1, from the European Space Agency (ESA; Torres et al. 2012), TerraSAR-X, from the German Aerospace Center (DLR; Werninghaus and Buckreuss 2009), CosmoSkyMed, from the Italian Space Agency (ASI; Covello et al. 2010), RADARSAT-2, from the Canadian Space Agency (CSA; Morena et al. 2004), and ALOS-2/PALSAR-2, from the Japan Aerospace Exploration Agency (JAXA; Kankaku et al. 2013). Among these, only Sentinel-1 provides public access to SAR data that can be used to estimate flooding.

The nature of LEO satellites makes SAR data sparse, however, and, unlike with precipitation products, gaps cannot be filled using geosynchronous satellites. In fact, the spatial resolution needed for accurate flooding maps is three orders of magnitude higher than that required for a global precipitation product. Even when observations are present, no detailed processing methods exist for real-time retrieval of flooding data because underdetection or overdetection issues necessitate manual labor (Shen et al. 2019a).

Recently, we published the RAdar-Produced Inundation Diary (RAPID) NRT fully automated system (Shen et al. 2019a), which involves the processing of high-resolution (10 meters) SAR images to allow the creation of rapid and efficient flood inundation maps by addressing both underdetection and overdetection. Differently than optical sensors, SAR images are nearly not affected by adverse weather conditions. As discussed in the methodology section below, the system is triggered (Yang et al. 2019) by IMERG precipitation estimates (Huffman et al. 2019), and it processes Sentinel-1 SAR data.

In this brief communication, we present the early results we delivered with the RAPID NRT automated system two days after Hurricane Dorian hit the Bahamas, just a few hours after SAR data for the event became publicly available. We provide a short description of the methodology, and we detail the extent of flood inundation by analyzing RAPID maps.

## 2 Methodology

Only a few SAR-based flood delineation methods (e.g. Horritt et al. 2003, Martinis et al. 2009, Matgen et al., 2011, Giustarini et al. 2012, Lu et al. 2014, Chini et al. 2017, Cian et al. 2018) have the potential to be fully automated (Shen et al. 2019b). The RAPID algorithm (Shen et al. 2019a) is an automated system capable of producing NRT inundation maps by processing SAR observations. Because of the considerable computation, storage, and data transfer time that would be needed to run the RAPID algorithm blindly for every SAR image worldwide, we implemented a zoom-in triggering mechanism (Yang et al. 2019) that allows selection of areas of the world where flooding is possible. Areas are defined by the availability of SAR images associated with land that has received at least 60 mm of accumulated precipitation during the previous day or potentially fluvial flood areas indicated by hydrological station observations. Within the continental United States (CONUS) area, we use the National Oceanic and Atmospheric Administration (NOAA) NEXt-Generation RADar (NEXRAD) precipitation product (NOAA 1991) and the U.S. Geological Survey (USGS) WaterWatch (https://waterwatch.usgs.gov/). We use NASA's IMERG, version 06 (Huffman et al. 2019), for the rest of the world (Yang et al. 2019).

After being triggered, the RAPID core algorithm (Shen et al. 2019a) handles both polarizations of SAR images in Ground Range Detected (GRD) mode through four steps: (1) identification of water and land pixels through a binary classification; (2) selection of water pixels connected to known water bodies and water areas not connected to known water bodies; (3) generation of a buffer region around the identified water bodies to reduce false negatives using less restrictive thresholds derived from the radar noise model; and (4) correction of the classification through a machine-learning algorithm that uses high-resolution topography (Farr et al. 2007), hydrography (Yamazaki et al. 2019), water occurrence (Pekel et al. 2016), and river bathymetry (Allen and Pavelsky 2018, Chen et al. 2019, Yamazaki et al. 2014). In step (2), the noise-reduced persistent water extent (know water body) is computed using at least 5 overpasses acquired during non-flood conditions for each pixel.

The RAPID system has been quantitatively compared in past studies with manually derived flood maps using (overall, user, producer) agreement scores, representing (accuracy, true positive rate, precision) parameters of the confusion matrix. Specifically, for Hurricane Harvey, RAPID was compared with the Dartmouth Flood Observatory (DFO) comprehensive flood map of August 30, 2017 (Shen et al., 2019) and against the USGS Dynamic Surface Water Extent (DSWE) Northwestern flood map of June 25, 2019 (Yang et al, 2019). RAPID yielded consistently high agreement scores for Harvey (93%, 75%, 77%) and the Northwestern flood (96%, 84%, 76%). For Hurricane Dorian, we are presenting a comparison between RAPID and the Copernicus Emergency Management Service (Copernicus EMS) first estimate maps (available at https://emergency.copernicus.eu/mapping/list-of-components/EMSR385/FEP/ALL), both derived from the Sentinel-1 SAR observations. Copernicus EMS flooding maps are not available for the entire SAR images, but only for the Abaco Islands on September 2, 2019, and for Grand Bahama on September 4, 2019.

## 3 Results

Because of the extremely high amounts of precipitation related to Hurricane Dorian (up to more than 1,400 mm over three days; see figure 1), the RAPID system was automatically triggered for the northern Bahamas. Sentinel-1 SAR data were available at 23:44 UTC on September 2 and at 11:09 UTC on September 4, 2019.

At the time of the first overpass on September 2, Dorian was located 20 km to the north of Grand Bahama (figure 1). Sentinel-1 data covered the northeastern sector of Grand Bahama and all of Great Abaco, both located in the southeastern sector of the hurricane and therefore affected by southwesterly winds. Lower-elevation areas on the west coast of Great Abaco were suffering onshore tropical storm force winds, and flooding in these territories was extensive, covering 518 km$^2$ of land, or 26% of the island (figure 2a).

On September 2, offshore hurricane-force winds affected lower-elevation areas along the northern coast of Grand Bahama, which had been affected by onshore hurricane-force winds during the previous day. Despite the blowing of the winds away from the coast, proximity to the center of the hurricane did not allow the storm surge to retreat significantly. For this reason, these locations were also still experiencing extensive flooding: 138 km$^2$ of the 308 km$^2$ covered by the SAR images were flooded, amounting to 45% of eastern Grand Bahama (figure 2a). Since the first overpass occurred several hours after the passage of the hurricane, the flooded area shown in figure 2a represents a conservative estimate.

The second Sentinel-1 overpass entirely captured Great Abaco, Grand Bahama, Andros, New Providence, and other smaller islands of the archipelago on September 4, when Dorian was located 300 km to the north of Grand Bahama. Despite the absence of storm surge at the time of the overpass, 17% of Great Abaco was still flooded, while flooding on Grand Bahama had mostly receded (14% of the island was still flooded).

On September 3, high amounts of precipitation also fell on Andros Island, located approximately 200 km to the south of Dorian's path. The flooding map resulting from the automated trigger of the RAPID algorithm also showed extensive flooding on the low-lying terrain of this island, which received onshore winds during the entire duration of the event (figure 2b). On September 4, the inundated area was 2,193 km$^2$ (37% of the island), and flooding reached more than 10 km inland. RAPID flooding estimates of area and inland extent on the Andros Island are in agreement with the coarser resolution product composited from the passive radiometers Visible Infrared Imaging Radiometer Suite (VIIRS) at 375m pixel spacing and the Advanced Baseline Imager (ABI) at 1km , displayed on the International Charter "Space and Major Disasters" website at https://disasterscharter.org/image/journal/article.jpg?img_id=3519568&t=1568272371731.

The agreement (overall, user, producer) scores between RAPID and Copernicus EMS flooding maps for the Abaco Islands on September 2 and September 4, derived from the confusion matrix shown in Table 1, were (77%, 90%, 41%) and (89%, 61%, 86%), respectively. The high overall and user agreement scores for the September 2 flooding are also depicted in the flood maps of Figure 2a indicating a very good overlap of the two products over the coast of Great Abaco, while the relatively low producer agreement comes from the lack of flood detection by the Copernicus EMS algorithm over the multiple near-sea-surface-elevation islands, located in the front of the western coast of Great Abaco. The relatively low user agreement score between the two products on September 4 is due to the fact that RAPID classifies some non-flooded areas within the Copernicus EMS flooded boundary, which are expected to occur as a consequence of the flood recession.

According to the conservative flooding estimates shown in these maps, the total area covered by flooding in the Bahamas exceeded 3,000 km$^2$, spread over areas hundreds of kilometers away from each other. To assess the inundation extent over such vast and dispersed areas, recognition flights take days, and they cannot operate during such extreme weather conditions as were presented by the long-lasting hurricane-force winds in the Bahamas. In contrast, a system such as RAPID can provide flooding estimates for any area of the world within hours of the data's becoming available. RAPID has the potential to be a fundamental tool for a fast and efficient emergency response.

## 4 Closing Remarks

Hurricane Dorian heavily damaged the northern Bahamas with extreme winds and precipitation and extensive flooding. When large-scale weather-related devastation occurs in areas of the world that do not have in situ observation networks, an assessment of the situation based on hydrometeorological parameters is often difficult.

In this brief communication, we analyzed the flooding related to Hurricane Dorian in the Bahamas at 10 m pixel spacing through RAPID, which is an automated system producing near-real-time flood maps across the globe based on SAR images. Specifically, RAPID identifies possibly flooded areas using near-real-time high-resolution precipitation products and then processes SAR images to compute inundation maps.

For Hurricane Dorian, RAPID inundation maps showed that, several hours after the passage of the storm, 26% of Great Abaco, 45% of eastern Grand Bahama, and 37% of Andros were flooded, for a total area exceeding 3,000 km$^2$. We compared RAPID inundation maps with Copernicus EMS maps, both obtained from freely available, very high resolution ESA Sentinel-1 SAR observations, finding high agreement scores and we discussed the differences between the two products for the case in exam. We believe RAPID system's ability to map such a large area of inundation as soon as SAR observations were available makes it a fast fully automated method for consistently assessing flood extension and providing situational awareness.

The main limitation of the system is the occasional unavailability of timely satellite overpasses in conjunction with heavy precipitation events. For Hurricane Dorian, Sentinel-1 images were not available at the peak of the event in the most affected area. This limitation can be overcome through international collaborations, such as the International Charter "Space and Major Disasters", Sentinel Asia, NASA-ISRO SAR Mission, and Copernicus Emergency Management Service – Mapping, that would increase the availability of data from other satellite missions.

Future extensions of this work will allow us to combine the rapidly derived inundated areas with high-resolution terrain elevation to identify flood levels and inversely estimate the surges that caused the flooding. Using this information, we would be able to extend the flood inundation estimates outside the SAR coverage, e.g. derive the September 2 flooding over northwestern part of Grand Bahama, where Sentinel-1 observations are not available. Furthermore, an estimate of the surge level can be valuable information for comparing with model forecasts for this event.

**Author contribution:** DC: conceptualization, supervision, writing – original draft. QY and MK: software, formal analysis, data curation. XS and EA: conceptualization, project administration, writing – review and editing.

**Competing interests:** The authors declare that they have no conflict of interest.

**Acknowledgements:** The authors of this publication had research support from Eversource Energy. The authors would like to thank the European Space Agency for granting free access to Sentinel-1 synthetic aperture radar data at 10m grid spacing,

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

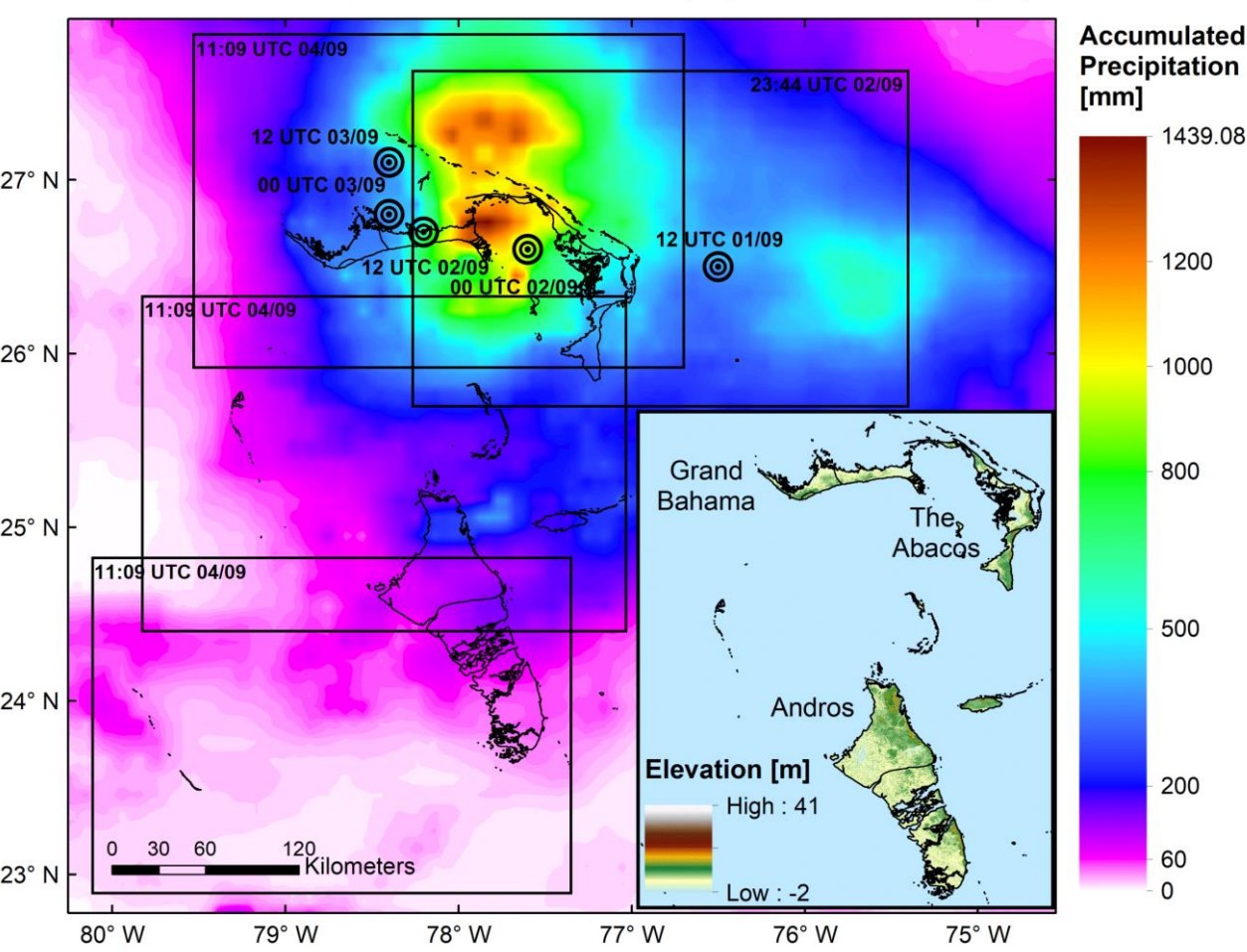

**Figure 1: Background map:** IMERG total accumulated precipitation between 00 UTC September 1 and 00 UTC September 4, 2019 (shaded), boundaries of Sentinel-1 SAR images (rectangles), and location (targets) of Hurricane Dorian from National Hurricane Center public advisories (NHC, 2019). Bottom right front map: elevation (Farr et al. 2007) of the study area. In both maps, Bahamas boundaries are delineated using the Global Administrative Areas dataset (Hijmans et al. 2015).

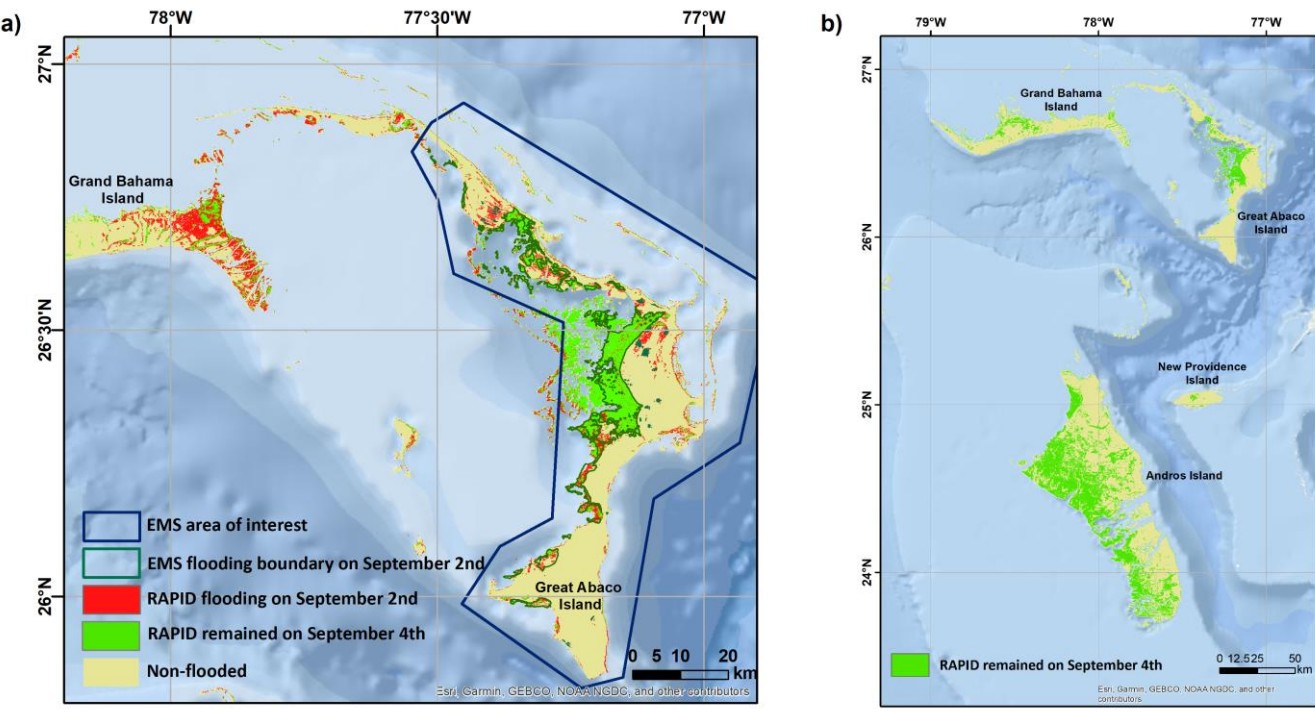

Figure 2: a) Flooded and non-flooded areas on September 2 and September 4, 2019 derived from the RAPID algorithm that processed SAR data from the Sentinel-1 overpasses, and flooded boundary on September 2 from Copernicus EMS. b) RAPID flooding map for the entire Northern and Central Bahamas, for the September 4, 2019 Sentinel-1 overpass. For both images, ocean background from World Ocean Base map (ESRI et al. 2014; list of contributors available at: http://downloads.esri.com/esri_content_doc/da/WorldOcean_ContributorsDA64.pdf).

| Confusion Matrix | | September 2 – Great Abaco | | September 4 – Grand Bahama | |
| --- | --- | --- | --- | --- | --- |
| | | Copernicus EMS | | Copernicus EMS | |
| | | Flooded | Non-flooded | Flooded | Non-flooded |
| RAPID | Flooded | 2,274,927 (**14.5%**) | 3,318,143 (**21.1%**) | 1,880,609 (**13.2%**) | 32,989 (**2.3%**) |
| | Non-flooded | 260,335 (**1.7%**) | 9,847,017 (**62.7%**) | 1,219,786 (**8.6%**) | 10,710,519 (**75.9%**) |

Table 1: Confusion matrix between RAPID and Copernicus EMS flooding products for September 2, 2019 overpass over Great Abaco (left) and for September 4, 2019 overpass over Grand Bahama (right). For each matrix, number and percentage of pixels is reported.