# Peer review of "Hurricane Dorian: automated near-real-time mapping of the “unprecedented” flooding on the Bahamas using SAR"

_Natural Hazards and Earth System Sciences, 2019_

## Referee Comment (RC1) · Anonymous Referee #1 · 12 Nov 2019

General comments

In this brief communication the authors present the results of a Sentinel-1-based automated near-real time flood mapping approach applied for detecting inundations caused in the frame of Hurricane Dorian on the Bahamas. Even if the inundations on the Bahamas were huge and very impressing the most important limitations of the work are in my view 1) the sole application of an already reported approach to a small subset of existing data sets which had been acquired over the Bahamas during this event, 2) the missing validation of the flood mapping results and 3) the lack of consideration of the work of other authors which achieved significant progress in SAR-based flood mapping

within the last years.

Specific comments - Abstract: In my view the focus of this contribution is not clear. The method for detecting flooding based on SAR data is already published by the authors and, as this is a brief communication, there is of course only shortly reported on the details of the methodology. Therefore, the focus of this publication should be on the huge flood event on the Bahamas. However, only Sentinel-1 data on two dates in early September has be analysed. By integrating other Earth Observation data sets acquired during this event (e.g. in the frame of the International Charter Space and Major Disasters) and also additional Sentinel-1 data acquired in September 2019 (e.g. on September 14) the evolution of this flood event could be better described (the RAPID approach could be of course a component to complete the description of this event).

- Line 38: Please replace Alos-2 by ALOS-2/PALSAR-2

- Line 43: There exist several automatic approaches/complete processing chains for detecting flood extent from different kind of radar satellite data (e.g. from TerraSAR-X, Sentinel-1, CosmoSkyMed). Multiple references have been published related to this topic within the last years by different organisations. Some of these references should be cited in this publication.

- Line 54: It would be better to cite directly the references related to automated flood delineation and not to refer only to previous work of the authors (Shen et al. 2019b)

- Line 65: It should be at least mentioned which Sentinel-1 data type (GRD or SLC) and polarization is used for extracting the flooding

- Result section: It would be important to perform an accuracy assessment of the flood masks

- Figure 2 and 3: it would be important to describe which data source was used to separate between normal water conditions and flooding. It would be helpful to visualize layers of normal water extent in the figures.

- Line 113: Without any information about the performance of RAPID and without any reference to other approaches in flood mapping reported in the literature I would suggest to remove the sentence: "We believe its ability to map such a large area of inundation so quickly makes RAPID the fastest fully automated method for assessing flood extension..."

- Line 115: These international collaborations or mechanisms exist and the authors should refer to them (e.g. International Charter "Space and Major Disasters", Sentinel Asia, Copernicus Emergency Management Service - Mapping).

---

## Author Comment (AC1) · 20 Dec 2019

**Reply to Reviewer Comments**

**General comments**

In this brief communication the authors present the results of a Sentinel-1-based automated near-real time flood mapping approach applied for detecting inundations caused in the frame of Hurricane Dorian on the Bahamas. Even if the inundations on the Bahamas were huge and very impressing the most important limitations of the work are in my view 1) the sole application of an already reported approach to a small subset of existing data sets which had been acquired over the Bahamas during this event, 2) the missing validation of the flood mapping results and 3) the lack of consideration of the work of other authors which achieved significant progress in SAR-based flood mapping within the last years.

We would like to thank the reviewer for his/her constructive comments. We have revised the paper according to all the reviewer's comments, and we are detailing our response (in green) in this document.

1) The reviewer is correct in stating that this paper discusses about a sole application of an already reported approach. However, this is the first times that this algorithm has been used in near-real time during an event, being now fully-automated. For these reasons, we chose the "Brief Communication" format to timely "disseminate information and data on topical events of significant scientific and/or social interest within the scope of the journal". In order to publish timely results, we did not test the case using all existing methods and we will argue in a following response that it is unnecessary to do so.

2) The RAPID system has been quantitatively validated using different methods by (Shen et al. 2019a) and (Yang et al. 2019) in different events and locations against different reference datasets. In the revised version of the manuscript we are reporting the accuracy values from the validation performed in those studies. Since RAPID has been thoroughly validated by other studies, this paper is a brief communication about the Bahamas inundation instead of the system development. Furthermore, we cannot find an independent third-party reference flood map extend to validate the estimates from the RAPID system. There are flood maps for September 2 and September 4, 2019 derived by ESA/EMS, but those are based on the same SAR observations. We included, in the new version of our manuscript, a comparison with these EMS maps.

3) Thank you for raising this point. The work of other authors had been extensively discussed in (Shen et al., 2019a) and (Shen et al., 2019b). Unfortunately, due to the nature of this paper, it cannot contain an extensive literature review. In the previous version of this manuscript we chose to cite only all the data sources used, and the work at the basis of this manuscript. In the revised version we also include some of the most important papers written by other authors: *"Only a few SAR-based flood delineation methods (e.g. Horritt et al. 2003, Martinis et al. 2009, Matgen et al., 2011, Giustarini et al. 2012, Lu et al. 2014, Chini et al. 2017, Cian et al. 2018) have the potential to be fully automated (Shen et al. 2019b)."* However, we would like to point out that although many algorithms claim a certain level of automation, they are limited in applications. For example, some method require bimodal histogram, while other are based on simple threshold-based methods suffering by over-detection of artificial surfaces. Most methods cannot directly be applied without human interference. The RAPID system overcomes all these issues and is the only system that requires no human interference from the identification of potential flooded areas to the final generation of flood maps from SAR. We retrospectively outputted all Sentinel-1 captured flood events in the CONUS area from 2016 to 2019, which is the main contribution of (Yang et al. 2019). Since the RAPID system, being fully automated, is more automated than other existing system, we 1) were the first to release the Bahamas flood maps to the public (half day

earlier than Copernicus/EMS), and 2) did not only map selected events of high impact or the peak of the selected event, but also reported every captured event in the CONUS area and every captured day in an event.

**Specific comments**

1. Abstract: In my view the focus of this contribution is not clear. The method for detecting flooding based on SAR data is already published by the authors and, as this is a brief communication, there is of course only shortly reported on the details of the methodology. Therefore, the focus of this publication should be on the huge flood event on the Bahamas. However, only Sentinel-1 data on two dates in early September has be analysed. By integrating other Earth Observation data sets acquired during this event (e.g. in the frame of the International Charter Space and Major Disasters) and also additional Sentinel-1 data acquired in September 2019 (e.g. on September 14) the evolution of this flood event could be better described (the RAPID approach could be of course a component to complete the description of this event).

Thank you for this comment. We agree with the reviewer about the clarity of the focus of our paper. In particular, we are not presenting the methodology for an automated system, but we are presenting the application of that system. For this reason we modified the following sentence of the abstract by adding the words "an application of":

*"we present an application of the automated near-real-time (NRT) system called RAdar-Produced Inundation Diary (RAPID) to European Space Agency Sentinel-1 SAR images to produce flooding maps for Hurricane Dorian in the northern Bahamas."*

The reviewer's question related to other Earth Observation data may arise by our lack of specificity on the resolution. In the previous version of the manuscript we generically wrote about "high-resolution". However, the resolution of this product is much higher than other high-resolution products: it is 10 meters (we included this information in the current version). Most of the other products are either at a lower resolution, or optical. Optical sensors do not work in adverse weather conditions are not reliable for an immediate response to hurricanes. To make this difference clear, we added the following sentence in the paper:

*"Differently than optical sensors, SAR images are not influenced by adverse weather conditions."*

Moreover, in the revised version we now include a comparison with the EMS product (which is based on the same SAR observation) in Figure 2, in the new Table 1, and in text:

*"The agreement (overall, user, producer) scores between RAPID and EMS flooding maps for the Abaco Islands on September 2 and September 4, derived from the confusion matrix shown in Table 1, were (77%, 90%, 41%) and (89%, 61%, 86%), respectively. The high overall and user agreement scores for the September 2 flooding are also depicted in the flood maps of Figure 2 indicating a very good overlap of the two products over the coast of Great Abaco, while the relatively low producer agreement comes from the lack of flood detection by the EMS algorithm over the multiple near-sea-surface-elevation islands, located in the front of the western coast of Great Abaco. The relatively low user agreement score between the two products on September 4 is due to the fact that RAPID classifies some non-flooded areas within the EMS flooded boundary, which are expected to occur as a consequence of the flood recession."*

[Figure]

**Figure 2:** Ocean background from World Ocean Base map (ESRI et al. 2014; list of contributors available at: http://downloads.esri.com/esri_content_doc/da/WorldOcean_ContributorsDA64.pdf).  Flooded and non-flooded areas on September 2 and September 4, 2019 derived from the RAPID algorithm that processed SAR data from the Sentinel-1 overpasses, and flooded boundary from EMS.

| Confusion Matrix | | September 2 – Great Abaco | | September 4 – Grand Bahama | |
|---|---|---|---|---|---|
| | | EMS | | EMS | |
| | | Flooded | Non-flooded | Flooded | Non-flooded |
| RAPID | Flooded | 2,274,927 (14.5%) | 3,318,143 (21.1%) | 1,880,609 (13.2%) | 32,989 (2.3%) |
| | Non-flooded | 260,335 (1.7%) | 9,847,017 (62.7%) | 1,219,786 (8.6%) | 10,710,519 (75.9%) |

**Table 1:** Confusion matrix between RAPID and EMS flooding products for September 2, 2019 overpass over Great Abaco (left) and for September 4, 2019 overpass over Grand Bahama (right). For each matrix, number and percentage of pixels is reported.

We also mentioned visible products available on the International Charter Space and Major Disasters website, which confirm the results we found for Andros Island:

*"RAPID flooding estimates of area and inland extent on the Andros Island are in agreement with the coarser resolution product composited from VIIRS (375m) and ABI (1km) passive radiometers, displayed on the International Charter "Space and Major Disasters" website at https://disasterscharter.org/image/journal/article.jpg?img_id=3519568&t=1568272371731."*

Finally, we would like to mention that the September 14 image, being acquired more than 10 days after the passage of the hurricane, shows just a very limited amount of flooded area. We believe it is not necessary to include this image in the brief communication.

[Figure]

Flooding remained on Sep.14th, Bahamas

2. Line 38: Please replace Alos-2 by ALOS-2/PALSAR-2

Thank you for this comment. We replaced Alos-2 with ALOS-2/PALSAR-2 in the revised manuscript.

3. Line 43: There exist several automatic approaches/complete processing chains for detecting flood extent from different kind of radar satellite data (e.g. from TerraSAR-X, Sentinel-1, CosmoSkyMed). Multiple references have been published related to this topic within the last years by different organisations. Some of these references should be cited in this publication.

The combined methods that can provide a complete processing for detecting flood extent from different kind of radar satellite data have been discussed in "Inundation Extent Mapping by Synthetic Aperture Radar: A Review" by Shen et al., 2019, which reads as follows:

*"Martinis et al. (2009) applied SBA –split-based approach- (Bovolo and Bruzzone 2007) to determine the global threshold for binary (water and non-water) classification. In SBA, a SAR image is first divided into splits (sub-tiles) to determine their individual thresholds using the Kittler and Illingworth (KI) method (Kittler and Illingworth 1986), global minimum, and quality index. Then, only qualified splits showing sufficient water and non-water pixels are selected to get the global threshold. The OO segmentation algorithm (implemented in e-cognition software) is used to segment the image into continuous and non-overlapping object patches at different scales. Then the global threshold is applied to each object. Eventually, topography is used as an option to fine-tune the results.*

*SBA is employed to deal with the heterogeneity of SAR backscattering from the same object in time and space. The intention of applying the OO segmentation algorithm is to reduce false alarms and speckle noise. OO was, however, originally designed for high-resolution optical sensors, which have no consideration of noise like speckle and water-like areas. The fine-tuning procedure can only deal with floodplain extended from identified water bodies, leaving inundated areas isolated from known water sources. To avoid the drawback of fixing tile size to SAR images of different places and resolution (Bovolo and Bruzzone 2007; Martinis et al. 2015; Martinis et al. 2009), Chini et al. (2017) propose the hierarchical SBA (HSBA) method with variable tile size, and they post-processed the binary water mask derived by HSBA using RGA and CD, similar to Giustarini et al. (2013); Matgen et al. (2011).*

*The ACM, also known as the snake algorithm (Horritt 1999), was, to the authors' knowledge, the first image segmentation algorithm designed for SAR data. It allows a certain amount of backscattering heterogeneity, while no smoothing across segment boundaries occurs. A smooth contour is favored by the inclusion of curvature and tension constraint. The algorithm spawns smaller snakes to represent multiple connected regions. The snake starts as a narrow strip moving along the course of a river channel, ensuring it contains only flooded pixels. Overall, it can deal with low signal to noise ratio.*

*Horritt et al. (2003) used ACM to map waterlines under vegetation. They started from known pure ocean pixels to map the active contour of open water and then to map the second active contour, which was the waterline beneath vegetation. Two radar signatures—the enhanced backscattering at C-band and the HH-VV phase difference at L-band—forced the ACM. Unlike the OO method, which aggregates objects from the bottom (pixel level) to the top, the segmentation in ACM requires seeding pixels, whose detection is difficult in an automated approach. In addition, similar to RGA, ACM cannot detect inundated areas isolated from a known water body."*

In our manuscript, we modified the first sentence in the Methodology section for including the above and additional references:

*"Only a few SAR-based flood delineation methods (e.g. Horritt et al. 2003, Martinis et al. 2009, Matgen et al., 2011, Giustarini et al. 2013, Lu et al. 2014, Chini et al. 2017, Cian et al. 2018) have the potential to be fully automated (Shen et al. 2019b)."*

However, we would like to mention that, to the authors' knowledge, Sentinel-1 is the only open data that is frequently availabe globally. In our paper we are only comparing Sentinel-1 based flood mapping results since satellites mentioned by the reviewer are not accessible by everyone including the authors.

4. Line 54: It would be better to cite directly the references related to automated flood delineation and not to refer only to previous work of the authors (Shen et al. 2019b)

Thank you for this comment. We are now citing directly the references related to automated flood delineation:

*"Only a few SAR-based flood delineation methods (e.g. Horritt et al. 2003, Martinis et al. 2009, Matgen et al., 2011, Giustarini et al. 2013, Lu et al. 2014, Chini et al. 2017, Cian et al. 2018) have the potential to be fully automated (Shen et al. 2019b)."*

Also in this case, for a detailed literature review, we are still referring to Shen et al. 2019b, which reads as follows:

*"Toward automation, Matgen et al. (2011) developed the M2a algorithm to determine the threshold that makes the non-water pixels (below the threshold) best fit a gamma distribution—a theoretical distribution of any given class in a SAR image. They then extended flooded areas using RGA from detected water pixels using a larger threshold—99 percentile of the "water" backscatter gamma distribution—arguing that flood maps resulting from region growing should include all "open water" pixels connected to the seeds. Then they applied a change detection technique to backscattering to reduce over-detection within the identified water bodies caused by water-like surfaces, as well as to remove permanent water pixels.*
*Based on the same concept, Giustarini et al. (2013) developed an iterative approach to calibrate the segmentation threshold, distribution parameter, and region growing threshold (M2b). They applied the same segmentation threshold to the dry reference SAR image to obtain the permanent water area. They claimed, however, that if the intensity distribution of the SAR image were not bimodal, the automated threshold determination might not work.*
*Lu et al. (2014) used a changed detection approach, first to detect a core flood area that contained a more plausible but incomplete collection of flood pixels, and then to derive the statistical curve of the water class to segment water pixels. The major advantage of this approach is that a bimodal distribution is not compulsory. In practice, a non-bimodal distribution often occurs. The change detection threshold might be difficult to determine and globalize.*
*Assuming even prior probability of flooded and non-flooded conditions, Giustarini et al. (2016) computed probabilistic flood maps that characterize the uncertainty of flood delineation. The probability reported in this study, however, related to the uncertainty neither in extent nor in time. Rather, it was the uncertainty of a SAR image classification based on backscattering.*
*Taking advantage of big earth observation (EO) data, the two most recent studies—Cian et al. (2018) and Shen et al. (2019)—implemented full automation of inundation retrieval. With the CD principle underpinning both methods, they employed multiple dry references instead of one supported by operational satellite SAR data for multiple years.*
*Cian et al. (2018) developed two CD-based flood indices, the Normalized Difference Flood Index (NDFI) and the Normalized Difference Vegetated Flood Index (NDVFI), assuming a number of revisits for each pixel in dry conditions was available."*

5. Line 65: It should be at least mentioned which Sentinel-1 data type (GRD or SLC) and polarization is used for extracting the flooding

Thank you. The Sentinel- data type is GRD and RAPID uses both channels in the dual polarization modes but could also work if occasionally single polarization or fully polarization data were provided. We added this information in the revised version:

*"the RAPID core algorithm (Shen et al. 2019a) handles both polarizations of SAR images in GRD mode through four steps".*

6. Result section: It would be important to perform an accuracy assessment of the flood masks

As responded in our item 2 to the general comments, for Bahamas, it is difficult to find another fully independent reference of flood extent at the same time, at a comparably high resolution, and covering the same area. However, the RAPID system has already been validated in two instances: for Hurricane Harvey and for the Northwestern Floods. In the current version of the manuscript, beyond mentioning these validations, we also included a comparison with EMS maps for the Bahamas. We included the details of the validation in the Methodology section:

*"The RAPID system has been quantitatively validated in past studies against manually derived flood maps using (overall, user, producer) agreement scores, representing (accuracy, true positive rate, precision) parameters of the confusion matrix. Specifically, for Hurricane Harvey, RAPID was validated against the DFO comprehensive flood map of August 30, 2017 (Shen et al., 2019) and against the USGS DSWE Northwestern flood map of June 25, 2019 (Yang et al, 2019). RAPID yielded consistently high agreement scores for Harvey (93%, 75%, 77%) and the Northwestern flood (96%, 84%, 76%). For Hurricane Dorian, we are presenting a comparison between RAPID and the Copernicus Emergency Management Service (EMS) first estimate maps (available at https://emergency.copernicus.eu/mapping/list-of-components/EMSR385/FEP/ALL), both derived from the Sentinel-1 SAR observations. EMS flooding maps are not available for the entire SAR images, but only for the Abaco Islands on September 2, 2019, and for Grand Bahama on September 4, 2019."*

We described the comparison between EMS and RAPID for both September 2 and September 4 in the Results section:

*"The agreement (overall, user, producer) scores between RAPID and EMS flooding maps for the Abaco Islands on September 2 and September 4, derived from the confusion matrix shown in Table 1, were (77%, 90%, 41%) and (89%, 61%, 86%), respectively. The high overall and user agreement scores for the September 2 flooding are also depicted in the flood maps of Figure 2 indicating a very good overlap of the two products over the coast of Great Abaco, while the relatively low producer agreement comes from the lack of flood detection by the EMS algorithm over the multiple near-sea-surface-elevation islands, located in the front of the western coast of Great Abaco. The relatively low user agreement score between the two products on September 4 is due to the fact that RAPID classifies some non-flooded areas within the EMS flooded boundary, which are expected to occur as a consequence of the flood recession."*

| Confusion Matrix | | September 2 – Great Abaco | | September 4 – Grand Bahama | |
| --- | --- | --- | --- | --- | --- |
| | | EMS | | EMS | |
| | | Flooded | Non-flooded | Flooded | Non-flooded |
| RAPID | Flooded | 2,274,927 (14.5%) | 3,318,143 (21.1%) | 1,880,609 (13.2%) | 32,989 (2.3%) |
| | Non-flooded | 260,335 (1.7%) | 9,847,017 (62.7%) | 1,219,786 (8.6%) | 10,710,519 (75.9%) |

**Table 2: Confusion matrix between RAPID and EMS flooding products for September 2, 2019 overpass over Great Abaco (left) and for September 4, 2019 overpass over Grand Bahama (right). For each matrix, number and percentage of pixels is reported.**

7. Figure 2 and 3: it would be important to describe which data source was used to separate between normal water conditions and flooding. It would be helpful to visualize layers of normal water extent in the figures.

   RAPID uses dry references for change detection. We vote (some studies name it the temporal filtering technique) each pixel using multiple dry references (no less than 5 overpasses) to create a noise-free persistent water extent (normal water extent).

   We added the following sentence in the Methodology section: *"In step 2, the noise-free persistent water extent (know water body) is computed using at least 5 dry overpasses for each pixel."*

8. Line 113: Without any information about the performance of RAPID and without any reference to other approaches in flood mapping reported in the literature I would suggest to remove the sentence: "We believe its ability to map such a large area of inundation so quickly makes RAPID the fastest fully automated method for assessing flood extension"

   Since in the revised version we are now providing the accuracy information of RAPID for different events, showing consistently high performance, we can safely make the conclusion. Other approaches have been extensively discussed in Shen et al., 2019a and, for brevity, cannot be discussed here.

9. Line 115: These international collaborations or mechanisms exist and the authors should refer to them (e.g. International Charter "Space and Major Disasters", Sentinel Asia, Copernicus Emergency Management Service - Mapping).

   Thank you for this comment. We included these collaborations in the revised version of our manuscript:

   *"This limitation can be overcome through international collaborations, such as the International Charter "Space and Major Disasters", Sentinel Asia, NASA-ISRO SAR Mission and Copernicus Emergency Management Service – Mapping, that may increase the availability of data from other satellite missions."*

   References:

   Bovolo, F., & Bruzzone, L. (2007). A split-based approach to unsupervised change detection in large-size multitemporal images: Application to tsunami-damage assessment. IEEE Transactions on Geoscience and Remote Sensing, 45, 1658-1670

   Chini, M., Hostache, R., Giustarini, L., Matgen, P.J.I.T.o.G., & Sensing, R. (2017). A hierarchical split-based approach for parametric thresholding of SAR images: Flood inundation as a test case, 55, 6975-6988

   Cian, F., Marconcini, M., & Ceccato, P. (2018). Normalized Difference Flood Index for rapid flood mapping: Taking advantage of EO big data. Remote Sensing of Environment, 209, 712-730

   Giustarini, L., Hostache, R., Kavetski, D., Chini, M., Corato, G., Schlaffer, S., Matgen, P.J.I.T.o.G., & Sensing, R. (2016). Probabilistic flood mapping using synthetic aperture radar data, 54, 6958-6969

   Giustarini, L., Hostache, R., Matgen, P., Schumann, G.J.-P., Bates, P.D., & Mason, D.C. (2012). A change detection approach to flood mapping in urban areas using TerraSAR-X. Geoscience and Remote Sensing, IEEE Transactions on, 51, 2417-2430

   Horritt, M. (1999). A statistical active contour model for SAR image segmentation. Image and Vision Computing, 17, 213-224

   Horritt, M.S., Mason, D.C., Cobby, D.M., Davenport, I.J., & Bates, P.D. (2003). Waterline mapping in flooded vegetation from airborne SAR imagery. Remote Sensing of Environment, 271–281

   Kittler, J., & Illingworth, J. (1986). Minimum error thresholding. Pattern recognition, 19, 41-47

Lu, J., Giustarini, L., Xiong, B., Zhao, L., Jiang, Y., & Kuang, G. (2014). Automated flood detection with improved robustness and efficiency using multi-temporal SAR data. Remote Sensing Letters, 5, 240-248

Martinis, S., Kersten, J., & Twele, A. (2015). A fully automated TerraSAR-X based flood service. ISPRS Journal of Photogrammetry and Remote Sensing, 104, 203-212

Martinis, S., Twele, A., & Voigt, S. (2009). Towards operational near real-time flood detection using a split-based automatic thresholding procedure on high resolution TerraSAR-X data. Natural Hazards and Earth System Sciences, 9, 303-314

Matgen, P., Hostache, R., Schumann, G., Pfister, L., Hoffmann, L., & Savenije, H. (2011). Towards an automated SAR-based flood monitoring system: Lessons learned from two case studies. Physics and Chemistry of the Earth, Parts A/B/C, 36, 241-252

Shen, X., Anagnostou, E.N., Allen, G.H., Brakenridge, G.R., & Kettner, A.J. (2019). Near Real-Time Nonobstructed Flood Inundation Mapping by Synthetic Aperture Radar. Remote Sensing of Environment, 221, 302-335

Yang, Q., Shen, X., Anagnostou, E.N., Mo, C., Eggleston, J.R., & Kettner, A.J. (2019). An Unprecedented High-Resolution Flood Inundation Archive (2016-present) from Sentinel-1 SAR imageries over the CONUS. Bulletin of American Meteorological Society, (proposal accepted)

---

## Referee Comment (RC2) · Guy J.-P. Schumann (Referee) · 22 Dec 2019

First of all, sorry I am a bit late with this and I hope my comments are still useful. I think they have also been highlighted in part and in other words by the other reviewer and the authors already posted some comments.

The case study presented is interesting but what is lacking is some detail on the actual data processed, some kind of validation (although difficult here) but then some cross-validation at least with other available maps processed either from the same Sentinel-1 by other organizations or maps processed from other EO imagery.

[Figure]

It would also be useful to put the processing and used processing chain in context with other fully automated methods used - there are now many of those, for example HASARD by LIST or the chain used at DLR ZKI or indeed maps from the Dartmouth Flood Observatory. Such maps can then also be used to cross-validate and get a good idea about the sensitivity/uncertainty in the presented flood maps.

I am not sure about the nature and scope of brief communications in NHESS but at this point the presented paper reads like a story or account of the event rather than a brief communication of science and assessment of results.

---

## Author Comment (AC2) · 5 Feb 2020

**Reply to Reviewer Comments**

**Comment:**

First of all, sorry I am a bit late with this and I hope my comments are still useful. I think they have also been highlighted in part and in other words by the other reviewer and the authors already posted some comments. The case study presented is interesting but what is lacking is some detail on the actual data processed, some kind of validation (although difficult here) but then some cross validation at least with other available maps processed either from the same Sentinel-1 by other organizations or maps processed from other EO imagery. It would also be useful to put the processing and used processing chain in context with other fully automated methods used - there are now many of those, for example HASARD by LIST or the chain used at DLR ZKI or indeed maps from the Dartmouth Flood Observatory. Such maps can then also be used to cross-validate and get a good idea about the sensitivity/uncertainty in the presented flood maps. I am not sure about the nature and scope of brief communications in NHESS but at this point the presented paper reads like a story or account of the event rather than a brief communication of science and assessment of results.

**Response:**

We would like to thank the reviewer for his/her constructive comments. We agree with the reviewer: despite being the RAPID algorithm fully validated for historical cases, validation (or more specifically cross-validation) for this event was missing in the original version of the manuscript. In the revised manuscript we are now including a comparison between the RAPID and the EMS product (which is based on the same SAR observation) in Figure 2, in the new Table 1, and in text, and discuss differences between the two products.

In particular, we added this paragraph at the end of the Methodology section:

*"The RAPID system has been quantitatively validated in past studies against manually derived flood maps using (overall, user, producer) agreement scores, representing (accuracy, true positive rate, precision) parameters of the confusion matrix. Specifically, for Hurricane Harvey, RAPID was validated against the DFO comprehensive flood map of August 30, 2017 (Shen et al., 2019) and against the USGS DSWE Northwestern flood map of June 25, 2019 (Yang et al, 2019). RAPID yielded consistently high agreement scores for Harvey (93%, 75%, 77%) and the Northwestern flood (96%, 84%, 76%). For Hurricane Dorian, we are presenting a comparison between RAPID and the Copernicus Emergency Management Service (EMS) first estimate maps (available at https://emergency.copernicus.eu/mapping/list-of-components/EMSR385/FEP/ALL), both derived from the Sentinel-1 SAR observations. EMS flooding maps are not available for the entire SAR images, but only for the Abaco Islands on September 2, 2019, and for Grand Bahama on September 4, 2019."*

We also added this paragraph towards the end of the Results section:

*"The agreement (overall, user, producer) scores between RAPID and EMS flooding maps for the Abaco Islands on September 2 and September 4, derived from the confusion matrix shown in Table 1, were (77%, 90%, 41%) and (89%, 61%, 86%), respectively. The high overall and user agreement scores for the September 2 flooding are also depicted in the flood maps of Figure 2 indicating a very good overlap of the two products over the coast of Great Abaco, while the relatively low producer agreement comes from the lack of flood detection by the EMS algorithm over the multiple near-sea-surface-elevation islands, located in the front of the western coast of Great Abaco. The relatively low user agreement score between the two products on September 4 is due to the fact that RAPID classifies some non-flooded areas within the EMS flooded boundary, which are expected to occur as a consequence of the flood recession."*

We replaced Figure 2 with the following:

[Figure]

**Figure 2:** Ocean background from World Ocean Base map (ESRI et al. 2014; list of contributors available at:
http://downloads.esri.com/esri_content_doc/da/WorldOcean_ContributorsDA64.pdf). Flooded and non-flooded areas on September 2 and September 4, 2019 derived from the RAPID algorithm that processed SAR data from the Sentinel-1 overpasses, and flooded boundary from EMS.

And we added Table 1:

| Confusion Matrix | | September 2 – Great Abaco | | September 4 – Grand Bahama | |
|---|---|---|---|---|---|
| | | EMS | | EMS | |
| | | Flooded | Non-flooded | Flooded | Non-flooded |
| RAPID | Flooded | 2,274,927 (14.5%) | 3,318,143 (21.1%) | 1,880,609 (13.2%) | 32,989 (2.3%) |
| | Non-flooded | 260,335 (1.7%) | 9,847,017 (62.7%) | 1,219,786 (8.6%) | 10,710,519 (75.9%) |

**Table 1:** Confusion matrix between RAPID and EMS flooding products for September 2, 2019 overpass over Great Abaco (left) and for September 4, 2019 overpass over Grand Bahama (right). For each matrix, number and percentage of pixels is reported.

Beyond the additions implemented in the paper, we also visually compared, in Figure (a), the VV-pol SAR images on July 4 (dry condition, left) and on September 2 2019 (peak flooding, right). This visual and subjective comparison will not be included in the paper due to space limitations. From the comparison, however, it is evident that all the flat islands in oval 1 in Figure (a), are flooded. Large areas in oval 2 (zoomed in the bottom images for facilitating the comparison) are also partially flooded, despite they are not as dark as other areas. The same occurs for the other red circles. These flooded or partially flooded areas are not captured by the EMS algorithm (Figure 2), probably due to the use of fixed

[Figure]

**Figure a:** VV-polarized SAR images (in dB scale) over Bahamas on July 4, 2019 (left column) and on September 2, 2019 (right column).

thresholds. In RAPID, the segmentation threshold is instead optimized individually for each image to reach the best goodness of fit of the theoretical water-class distribution.

In order to put the RAPID processing chain in context with other automated methods, we introduced the following sentence: *Only a few SAR-based flood delineation methods (e.g. Horritt et al. 2003, Martinis et al. 2009, Matgen et al., 2011, Giustarini et al. 2012, Lu et al. 2014, Chini et al. 2017, Cian et al. 2018) have the potential to be fully automated (Shen et al. 2019b)."* All these references have been extensively discussed in Shen et al. 2019b, which reads as follows:

*"Toward automation, Matgen et al. (2011) developed the M2a algorithm to determine the threshold that makes the non-water pixels (below the threshold) best fit a gamma distribution—a theoretical distribution of any given class in a SAR image. They then extended flooded areas using RGA from detected water pixels using a larger threshold—99 percentile of the "water" backscatter gamma distribution—arguing that flood maps resulting from region growing should include all "open water" pixels connected to the seeds. Then they applied a change detection technique to backscattering to reduce over-detection within the identified water bodies caused by water-like surfaces, as well as to remove permanent water pixels. Based on the same concept, Giustarini et al. (2013) developed an iterative approach to calibrate the segmentation threshold, distribution parameter, and region growing threshold (M2b). They applied the same segmentation threshold to the dry reference SAR image to obtain the permanent water area. They claimed, however, that if the intensity distribution of the SAR image were not bimodal, the automated threshold determination might not work.*
*Lu et al. (2014) used a changed detection approach, first to detect a core flood area that contained a more plausible but incomplete collection of flood pixels, and then to derive the statistical curve of the water class to segment water pixels. The major advantage of this approach is that a bimodal distribution is not compulsory. In practice, a non-bimodal distribution often occurs. The change detection threshold might be difficult to determine and globalize.*
*Assuming even prior probability of flooded and non-flooded conditions, Giustarini et al. (2016) computed probabilistic flood maps that characterize the uncertainty of flood delineation. The probability reported in this study, however, related to the uncertainty neither in extent nor in time. Rather, it was the uncertainty of a SAR image classification based on backscattering.*
*Taking advantage of big earth observation (EO) data, the two most recent studies—Cian et al. (2018) and Shen et al. (2019)—implemented full automation of inundation retrieval. With the CD principle underpinning both methods, they employed multiple dry references instead of one supported by operational satellite SAR data for multiple years.*
*Cian et al. (2018) developed two CD-based flood indices, the Normalized Difference Flood Index (NDFI) and the Normalized Difference Vegetated Flood Index (NDVFI), assuming a number of revisits for each pixel in dry conditions was available."*

About automation of the entire processing chain, on the LIST website, at https://www.list.lu/en/news/list-contributes-to-monitor-mozambique-floods-with-satellite-imagery/ is written that: *"The project partners intend to develop a fully automated tool - based on HASARD ® - that could generate different flood risk maps, with no human intervention, as soon as a flood disaster occurs."* Therefore, according to the information written on the website, HASARD triggering by LIST is currently not automated.

The DLR ZKI needs to be activated too, and activation for Hurricane Dorian does not appear on the website: https://activations.zki.dlr.de/viewer/#/en/georss

With the addition of the comparison between RAPID and EMS, the discussion of those results, and the addition of references related to other automated methods, we hope to have addressed the reviewer's concerns about the science scope of this brief communication.

---

## Author Response (AR1)

**Author's response and marked up version**

**Editor report:**

Thank you very much again for submitting your brief communication manuscript 'Hurricane Dorian: automated near-real-time mapping of the "unprecedented" flooding on the Bahamas using SAR'. Two referee reports are available with major comments raised regarding the scope of your work and the validation of the applied method, and accordingly major revisions are needed.

Reading your responses to these comments I am positive that a revised version of your manuscript will address these points appropriately.

As a next step, I kindly ask you to provide a revised marked up (track changes) version of your manuscript to make clear how you include the changes in response to the referee report. Please also take note of the specifications regarding pages, figures etc. for brief communication manuscripts (https://www.natural-hazards-and-earth-system-sciences.net/about/manuscript_types.html).

I look forward to receiving the revised version of your manuscript.

**Author's response:**

We appreciate the Editor's positive view on our responses to the referee reports.

We are providing, as requested, a marked up (track changes) version of our manuscript, in which the implementation of the changes made to address the reviewers' comments is marked in red.

In order to meet the brief communication specifications, we reduced the abstract length to 97 words, we merged Figure 2 and Figure 3 into a unique Figure 2 (since we added Table 1), we included the sections Author Contribution and Competing Interests. Given that reviewers asked us to add more references, we are asking flexibility on the number of references allowed (we currently include 29 references).

[revised manuscript text omitted]

---

## Referee Report (RR1)

**General comments**

There have been some improvements of the manuscript related to validation and the citation of state of the art methods. However, some of my comments of the first round of review have not been considered.

**Specific comments**

- In my view the focus of this contribution is still not clear. The method for detecting flooding based on SAR data is already published by the authors. Therefore, the focus of this publication should be on the huge flood event on the Bahamas. However, only Sentinel-1 data on two dates in early September has been analysed. By integrating other Earth Observation data sets acquired during this event (e.g. in the frame of the International Charter Space and Major Disasters) and also additional Sentinel-1 data acquired in September 2019 (e.g. on September 14) the evolution of this flood event could be better described (the RAPID approach could be of course a component to complete the description of this event) (see also my comment of my first review).
- Line 49: Replace (EMS) by (CMES)
- Line 74: Please replace Alos-2 by ALOS-2/PALSAR-2 (see also my comment in the first review)
- Line 83: I think 10 meters is the pixel spacing and not the spatial resolution. I would suggest to use the abbreviation "m" instead of "meters"
- Line 85: X-band data can be affected by adverse weather conditions. I would suggest writing "SAR images are nearly not affected…"
- Line 110: I do not understand the meaning of "dry overpasses". You mean acquired during dry conditions? Perhaps it is better to use the term "non-flood conditions"? What is the meaning of noise-free? Is this derived by combining the data of 5 overpasses? If yes, it would be perhaps better to replace this term by "noise-reduced".
- Line 113: Please specify "DFO" and "DSWE"
- Line 143: The comparison with Charter-based products is only conducted on a visual basis, correct?
- Line 147-154: In order to perform a validation, the validation data should be correct. If there are errors in the CMES products I would suggest not to use the data as basis for validation or just to perform a comparison between the results (and not a validation).
- Line 175: Without any information about the performance of RAPID I would strongly suggest to remove the sentence: "We believe RAPID system's ability to map such a large area of inundation as soon as SAR observations were available makes it the fastest fully automated method for assessing flood extension and providing situational awareness". It

would be better to prove this statement. This was also my comment in the first round of review.

- Figure 2 and 3: it would be important to describe which data source was used to separate between normal water conditions and flooding. It would be helpful to visualize layers of normal water extent in the figures. This was also my comment in the first round of review.

---

## Author Response (AR2)

**Authors' response**

**Editor:**

Thank you very much again for your responses and revisions of your manuscript. The referee reports for these major revisions are now available. Both of them agree that your manuscript has improved and that you have addressed most of the comments. However, some additional revisions are requested. Some of them refer to comments from the previous round of major revisions. I kindly ask you to take the specific comments of both reviewers into thorough consideration and rework the manuscript accordingly. Sharpening the focus of your paper is an important recommendation. Therefore, I decide on minor revisions. Please provide a revised marked up (track changes) version of your manuscript to make clear how you include the changes in response to the referee report. I look forward to receiving the revised version of your manuscript.

We would like to thank the Editor for his advice on how to improve our manuscript. Given Reviewer 1 comments, we believe that he/she had missed our response that was uploaded to the system as a separate document. Below we provide our specific response to the comments listed by both Reviewers and, using a smaller font size, a repetition of our previous response to comments, which we believe we already addressed.

**Reviewer 1**

There have been some improvements of the manuscript related to validation and the citation of state of the art methods. However, some of my comments of the first round of review have not been considered.

We wrote a nine-page Response to Reviewer 1 document, by extensively considering and addressing all the comments provided in the first round of reviews. We believe that Reviewer 1 missed our Response to his/her comments that was uploaded to the system as a separate document. In any case, below we provide our specific response to the comments listed in his/her second review.

**Specific comments**

- In my view the focus of this contribution is still not clear. The method for detecting flooding based on SAR data is already published by the authors. Therefore, the focus of this publication should be on the huge flood event on the Bahamas. However, only Sentinel-1 data on two dates in early September has been analysed. By integrating other Earth Observation data sets acquired during this event (e.g. in the frame of the International Charter Space and Major Disasters) and also additional Sentinel-1 data acquired in September 2019 (e.g. on September 14) the evolution of this flood event could be better described (the RAPID approach could be of course a component to complete the description of this event) (see also my comment of my first review).

We would like to iterate herein our response that was submitted to address this exact same comment during the first round of reviews.

Thank you for this comment. We agree with the reviewer about the clarity of the focus of our paper. In particular, we are not presenting the methodology for an automated system, but we are presenting the application of that system. For this reason we modified the following sentence of the abstract by adding the words "an application of":

40 *"we present an application of the automated near-real-time (NRT) system called RAdar-Produced Inundation Diary (RAPID) to European Space Agency Sentinel-1 SAR images to produce flooding maps for Hurricane Dorian in the northern Bahamas."*

The reviewer's question related to other Earth Observation data may arise by our lack of specificity on the resolution. In the previous version of the manuscript we generically wrote about "high-resolution". However, the resolution of this product is much higher than other high-resolution products: it is 10 meters (we included this information in the current version). Most of the other products are either at a lower resolution, or optical. Optical sensors do not work in adverse weather conditions are not reliable for an immediate response to hurricanes. To make this difference clear, we added the following sentence in the paper:

45 *"Differently than optical sensors, SAR images are not influenced* (more recent edit: nearly not affected) *by adverse weather conditions."*

Moreover, in the revised version we now include a comparison with the EMS product (which is based on the same SAR observation) in Figure 2, in the new Table 1, and in text:

*"The agreement (overall, user, producer) scores between RAPID and EMS flooding maps for the Abaco Islands on September 2 and September 4, derived from the confusion matrix shown in Table 1, were (77%, 90%, 41%) and (89%, 61%, 86%), respectively. The high overall and user agreement scores for the*
50 *September 2 flooding are also depicted in the flood maps of Figure 2 indicating a very good overlap of the two products over the coast of Great Abaco, while the relatively low producer agreement comes from the lack of flood detection by the EMS algorithm over the multiple near-sea-surface-elevation islands, located in the front of the western coast of Great Abaco. The relatively low user agreement score between the two products on September 4 is due to the fact that RAPID classifies some non-flooded areas within the EMS flooded boundary, which are expected to occur as a consequence of the flood recession."*

We also mentioned visible products available on the International Charter Space and Major Disasters website, which confirm the results we found for Andros
55 Island:

*"RAPID flooding estimates of area and inland extent on the Andros Island are in agreement with the coarser resolution product composited from VIIRS (375m) and ABI (1km) passive radiometers, displayed on the International Charter "Space and Major Disasters" website at https://disasterscharter.org/image/journal/article.jpg?img_id=3519568&t=1568272371731."*

Finally, we would like to mention that the September 14 image, being acquired more than 10 days after the passage of the hurricane, shows just a very limited
60 amount of flooded area. We believe it is not necessary to include this image in the brief communication.

- Line 49: Replace (EMS) by (CMES)

Thank you for this comment, we replaced "EMS" with "Copernicus EMS", which is the abbreviation used on the Copernicus Emergency Management System website.

65 - Line 74: Please replace Alos-2 by ALOS-2/PALSAR-2 (see also my comment in the first review)

Thank you for this comment. We apologize for having incorrectly written in the previous response that it has been replaced, when it actually has not been. Now we replaced Alos-2 with ALOS-2/PALSAR-2.

- Line 83: I think 10 meters is the pixel spacing and not the spatial resolution. I would suggest to use the abbreviation "m" instead of "meters"

70 Thank you, it is correct. 10 m is the pixel spacing, not the spatial resolution.

We replaced "meters" with "m" at line 46, and "10 meter resolution" with "10 m pixel spacing" at line 128.

- Line 85: X-band data can be affected by adverse weather conditions. I would suggest writing "SAR images are nearly not affected…"

Thank you. At line 48 of the new version of our manuscript we replaced the sentence, "SAR images are not
75 influenced by adverse weather conditions with the sentence "SAR images are nearly not affected by adverse weather conditions".

- Line 110: I do not understand the meaning of "dry overpasses". You mean acquired during dry conditions? Perhaps it is better to use the term "non-flood conditions"? What is the meaning of noise-free? Is this derived by combining the data of 5 overpasses? If yes, it would be perhaps better to replace this term by "noise-reduced".

80 Yes, we intend that we combine the data from 5 overpasses to reduce the noise. All 5 overpasses have been acquired during dry conditions.

We replaced "dry overpasses" with "overpasses acquired during non-flood conditions".

We also replaced "noise-free" with "noise-reduced".

- Line 113: Please specify "DFO" and "DSWE"

Thank you. We replaced "DFO" with "Dartmouth Flood Observatory (DFO)" and "DSWE" with "Dynamic Surface Water Extent (DSWE)"

- Line 143: The comparison with Charter-based products is only conducted on a visual basis, correct?

That's correct. There is a difference of two to three orders of magnitude in the pixel spacing.

- Line 147-154: In order to perform a validation, the validation data should be correct. If there are errors in the CMES products I would suggest not to use the data as basis for validation or just to perform a comparison between the results (and not a validation).

Thank you very much for this comment. We received the same comment from Reviewer 2 in this round of revisions. We replaced the words "validated…against" with "compared…with".

- Line 175: Without any information about the performance of RAPID I would strongly suggest to remove the sentence: "We believe RAPID system's ability to map such a large area of inundation as soon as SAR observations were available makes it the fastest fully automated method for assessing flood extension and providing situational awareness". It would be better to prove this statement. This was also my comment in the first round of review.

We already considered your comment in the first round of review, and we replied that:

*"Since in the revised version we are now providing the accuracy information of RAPID for different events, showing consistently high performance, we can safely make the conclusion. Other approaches have been extensively discussed in Shen et al., 2019a and, for brevity, cannot be discussed here."*

We hope that this argument would satisfy the reviewer as we truly believe that this statement should remain in the paper, because RAPID indeed is the fastest fully/automated method currently available to map flood inundation from SAR.

- Figure 2 and 3: it would be important to describe which data source was used to separate between normal water conditions and flooding. It would be helpful to visualize layers of normal water extent in the figures. This was also my comment in the first round of review.

We already considered your comment in the first round of review, and we replied that:

RAPID uses dry references for change detection. We vote (some studies name it the temporal filtering technique) each pixel using multiple dry references (no less than 5 overpasses) to create a noise-free persistent water extent (normal water extent).
We added the following sentence in the Methodology section: *"In step 2, the noise-free ((*more recent edit: noise-reduced*) persistent water extent (know water body) is computed using at least 5 dry overpasses (*more recent edit: overpasses acquired during non-flood conditions)* for each pixel."*

In this particular case showing an extra layer of the normal water extent is not very helpful, because that is the ocean.

**Reviewer 2:**

The authors have made efforts to address my previous concerns adequately, specifically, they added existing literature, references that describe the method used and most importantly they compared the results with other data sets. The only comment I have relates to the use of the term validation. Given that the datasets the authors use to "validate" their results have not been independently validated and thus no level of accuracy has been associated, I'd suggest the authors replace the word "validation"/"validate" with "comparison"/"compare".

**Response:**

We would like to thank Reviewer 2 for his comment.

We received the same comment from Reviewer 1 in this round of revisions. We replaced the words "validated…against" with "compared…with".

[revised manuscript text omitted]

---

## Author Response (AR3)

**Authors' Response**

**Editor:**

Dear Diego Cerrai and co-authors,

Thank you for providing your responses and revision of your manuscript.

Overall, I think you have addressed the remaining concerns of the referees and have given satisfactory replies. One exception is the statement in your conclusions in l 134 '...makes it the fastest fully automated method...' which I think is not fully supported by the results presented in your brief communication. I

have made a suggestion to relax this statement in the annotated PDF.

Further, reading through the manuscript I have spotted some small technical things which you can find in the annotated pdf-file. Please check if you can correct them.

*Once again, we would like to thank the Editor for contributing to improve this manuscript.*

*We changed the sentence in line 134, as suggested by the Editor, and addressed all the small remaining issues in the manuscript. We also replaced figure 2 with a new one which standardized the legend and the colors across Figure 2a and Figure 2b. We finally decided to remove the sentences on the death toll, since the current estimated amount may raise further due to long term effects (as it happened in Puerto Rico).*

*We are providing, in the rest of this document, a revised, marked up version of our manuscript.*

[revised manuscript text omitted]